# Association between the Preoperative Dietary Antioxidant Index and Postoperative Quality of Life in Patients with Esophageal Squamous Cell Carcinoma: A Prospective Study Based on the TTD Model

**DOI:** 10.3390/nu15132828

**Published:** 2023-06-21

**Authors:** Juwei Zhang, Jinsong Zhou, Yue Huang, Zheng Lin, Suhong Zhang, Minglian Qiu, Zhisheng Xiang, Zhijian Hu

**Affiliations:** 1Department of Epidemiology and Health Statistics, Fujian Provincial Key Laboratory of Environment Factors and Cancer, School of Public Health, Fujian Medical University, Fuzhou 350122, China; juweizhang@fjmu.edu.cn (J.Z.); jinsongzhou@fjmu.edu.com (J.Z.); huangyue@fjmu.edu.com (Y.H.); lz@fjmu.edu.com (Z.L.); suhongzhang@fjmu.edu.cn (S.Z.); 2Department of Thoracic Surgery, The First Affiliated Hospital of Fujian Medical University, Fuzhou 350004, China; qml-817@163.com; 3Department of Epidemiology, Clinical Oncology School of Fujian Medical University, Fujian Cancer Hospital, Fuzhou 350014, China; xzs650@126.com; 4Key Laboratory of the Ministry of Education for Gastrointestinal Cancer, Fujian Medical University, Fuzhou 350122, China

**Keywords:** esophageal squamous cell carcinoma, dietary antioxidant index, health-related quality of life, time to deterioration model

## Abstract

Objective: Dietary antioxidants are associated with risk of death in cancer patients, and they were used to evaluate the prognosis of cancer patients. Dietary antioxidant index (DAI) can be used to evaluate dietary antioxidant content comprehensively; this study aimed to investigate the effect of preoperative DAI on health-related quality of life in patients with esophageal cell squamous carcinoma (ESCC). Methods: Data on dietary intakes were collected using a validated food-frequency questionnaire (FFQ). DAI was calculated for all study participants based on FFQ data of each participant. The study involved conducting several follow-up activities with patients diagnosed with ESCC to evaluate their quality of life. The approach employed in the study was to conduct a telephone interview. The EORTC Quality of Life Questionnaire-Core Questionnaire (EORTC QLQ-C30, version 3.0) and the Esophageal Cancer Module (EORTC QLQ-OES18) were used to collect data on the quality of life of the patients; all patients completed the full follow-up. Results: This prospective study was performed on 376 participants who were recruited from Fujian Cancer Hospital and First Hospital of Fujian Medical University. They all were diagnosed with ESCC. The results indicated that the time to deterioration of global health status (*p* = 0.043), cognitive functioning (*p* = 0.031), dry mouth (*p* = 0.019), and speech problems (*p* = 0.031) significantly delay in the high DAI group. Univariate and multivariate Cox regression analysis showed that global health status (*HR* = 0.718, *95% CI*: 0.532–0.969), cognitive functioning (*HR* = 0.641, *95% CI*: 0.450–0.913), dry mouth (*HR* = 0.637, *95% CI*: 0.445–0.911), and speech problems (*HR* = 0.651, *95% CI*: 0.449–0.945) were improved in the high DAI group. Conclusions: Prognostic value of preoperative DAI was significant for patients with ESCC who undergo surgical intervention. Its level was positively correlated with the postoperative quality of life of patients, which can delay and improve the occurrence of postoperative physical function and symptom deterioration.

## 1. Introduction

Esophageal cancer is considered to be one of the most aggressive types of gastrointestinal cancers [1]. The global cancer statistics for 2020 indicate that there are 604,000 new and 544,000 deaths from esophageal cancer, making it the sixth leading cause of death worldwide [2]. Esophageal squamous cell carcinoma and adenocarcinoma are the main histological types, with ESCC being more common in China [3]. With the development of medical technology, radical esophagectomy and adjuvant treatment greatly improved the survival rate of patients with esophageal cancer [4]. However, esophagectomy has the potential to negatively impact quality of life due to its highly invasive nature [1], the postoperative enhancement of the quality of life for patients diagnosed with ESCC aroused widespread concern among researchers.

Health-related quality of life (HRQoL) can be defined as a “multidimensional concept that includes subjective reports of symptoms, side effects, functioning in multiple life domains, and general perceptions of life satisfaction and quality” by the US Food and Drug Administration (FDA) [5]. Despite the advances in precision medicine and personalized treatment, the ultimate objective for patients with cancer and healthcare providers is to achieve a cure. For numerous patients, the level of well-being experienced during and after treatment holds equal significance. Therefore, compared to the past, apart from the overarching objective of improving patients’ longevity and well-being, there is a growing recognition that HRQOL plays a vital role in determining the overall outcomes for patients [6]. Surgery can significantly impact the quality of life of patients due to the esophagus’ unique location, resulting in side effects like reflux and difficulty swallowing [7]. Therefore, finding factors to improve postoperative quality of life is of great significance for ESCC patients.

Dietary factors are closely related to the occurrence and development of esophageal squamous cell carcinoma. Fruits and vegetables contain various natural pigments [8], which have strong antioxidant activity and are negatively correlated with the occurrence of ESCC [9]. However, there is no consensus on the association between dietary antioxidant intake and cancer prognosis. The findings of a systematic review and meta-analysis [10] indicated that the oral antioxidants during treatment was not related to the survival rate of breast cancer. The author speculated that with chemotherapy and radiation therapy producing reactive oxygen species (ROS) as one of their anti-tumor mechanisms, excessive dietary antioxidants may neutralize ROS [11]. This result addressed concerns that supplementing too many antioxidant nutrients during the treatment process will not reduce treatment effectiveness. In contrast, another cohort study [12] proved that prior consumption of antioxidants could lead to an improved survival rate among patients diagnosed with breast cancer, while the intake during treatment was not related to the survival rate of breast cancer patients. The above research suggests that the impact of dietary antioxidants on cancer prognosis may be closely related to the intake stage, and intake before diagnosis has a positive impact on prognosis. To date, there are a lack of studies investigating the potential relationship between pre-diagnostic intake of dietary antioxidants and postoperative quality of life in patients with ESCC.

Dietary intake of antioxidants is a complex system. A diverse range of antioxidants, such as catechins, flavonoids, anthocyanins, stilbenes, and carotenoids, are present in the diet [13], Wright’s study [14] recommended a nutritional assessment tool, the dietary antioxidant index (DAI) [15], also known as the composite dietary antioxidant index (CDAI), including multiple dietary antioxidants such as manganese, zinc, selenium, and vitamins A, C, and E, which assess dietary intake of a combination of antioxidants. The DAI index was developed by taking markers such as tumor necrosis factor-α and interleukin-1β into consideration and evaluating the combined anti-inflammatory effects of various foods [16]. Moreover, due to their crucial function in the majority of diets globally, there is growing attention towards exploring the potential health benefits associated with consuming foods that are high in total antioxidant capacity (TAC). Previous studies found a negative correlation between DAI and cancer risk, such as gastric cancer [17], breast cancer [18], and lung cancer [14]. However, similar studies were not yet conducted on the prognosis of cancer patients.

In this paper, we investigated the effect of preoperative dietary antioxidant index on health-related quality of life in patients with ESCC. Based on our knowledge, this study is the pioneering research effort to examine the potential association between preoperative consumption of dietary antioxidants and the quality of life post-surgery among individuals suffering from ESCC. The purpose of this study was to provide scientific suggestions for the prevention and treatment of ESCC.

## 2. Methods

### 2.1. Study Design and Participants

This prospective study was performed on 376 participants who were recruited from Fujian Cancer Hospital and First Hospital of Fujian Medical University from December 2014 to November 2021. The aim of the study was to evaluate the quality of life in patients who were diagnosed with esophageal squamous cell carcinoma (ESCC) and were invited to participate. Specific inclusion and exclusion methods refer to our previous studies [19]. Tumor staging was established according to the criteria outlined by the American Joint Committee on Cancer Tumor Lymph Node Metastasis (TNM) classification system. Approval for this study (approval number: 201495) was obtained from the Ethics Committee of Fujian Medical University, and all participants provided informed consent prior to their involvement in the research.

### 2.2. Health-Related Quality

The standardized questionnaire used in this study were the EORTC Quality of Life Questionnaire-Core Questionnaire (EORTC QLQ-C30, version 3.0) [20] and the Esophageal Cancer Module (EORTC QLQ-OES18) [21], which was administered face-to-face by trained interviewers to all enrolled patients within 72 h of hospitalization. The detailed content of the questionnaire was described in our previous articles [19]. Postoperative patients were monitored to evaluate their HRQOL after undergoing esophagectomy. This involved follow-up assessments at intervals of 3 months during the first year and 6 months in the subsequent years. 

The time to deterioration (TTD) model was utilized to assess the effectiveness of HROQL. Specifically, the first instance of a 5-point drop in score for each domain from the initial evaluation before surgery was considered as the deterioration event. TTD was then defined as the duration between the initial evaluation and the point of deterioration or as the final follow-up timepoint for patients who did not experience any deterioration [22].

### 2.3. Questionnaire Assessment

The “Resident Health Status Questionnaire” was used in this study, face-to-face surveys were conducted by uniformly trained investigators to collect the general information (age, gender, ethnicity, education level, place of residence, marital status, height, weight, etc.), lifestyle habits (history of smoking, alcohol consumption, tea consumption, etc.), family history of tumors, history of previous diseases, occupational history, etc., of the study participants, which were completed within 3 days after admission.

A semi-quantitative food frequency questionnaire (FFQ) was used to investigate the frequency and intake of various foods in the past year in the study participants to obtain the long-term food consumption patterns and dietary habits of individuals. A total of 185 food items were included in 12 categories, including staple foods, potatoes, eggs, soy products, animal meat, fish, shrimp, crab and shellfish, vegetables, fruits, and dairy. For each food item, the number of times consumed per day/week/month/year, the amount of each intake and the cumulative duration of intake were used to investigate its intake. For those who did not know the food weight, the food model legend and food weight conversion table were referred to. Conversion of weekly food intake frequency: the food intake frequency was converted to “times/week” according to the equivalence formula (e.g., if eggs were consumed once a day, 1 time/day × 7 days/week = 7 times/week); finally, the weekly intake of each food was calculated (weekly intake = weekly intake frequency × each consumption) in grams (g) as unit. Exclusion criteria: (1) extreme daily intake (<1% or >99%) or extreme intake of any one food (>99.5%); (2) intake of less than 10 types of food. Finally, the daily intakes of six antioxidants, vitamins A, C, and E, manganese, selenium, and zinc, were calculated according to the Chinese Food Composition Table (6th edition) based on the nutrient composition of each food.

### 2.4. Dietary Antioxidant Index (DAI)

DAI was calculated for all study participants based on FFQ data. Wright’s recommended method was utilized to measure the total antioxidant capacity of the food consumed by the participants [14]. For each participant, DAI was determined by adding together the standardized intake of six different antioxidants (i): vitamins A, C, and E, as well as manganese, selenium, and zinc, which were derived from food sources only (i.e., excluding dietary supplements). This is described in the formula below. In the equation, x_i_ is the daily intake of antioxidant; μ_i_ is the mean value of antioxidant i throughout the overall study; and S_i_ is the SD of μ.
$$\mathrm{DAI}=\overset{n=6}{\underset{i=1}{\sum }}\frac{X_{i}-\mu_{i}}{S_{i}}$$

ESCC patients are divided into low group (DAI < 0.729) and high group (DAI ≥ 0.729) based on the median DAI.

### 2.5. Statistical Methods

The chi-square test was utilized to compare the distribution of categorical variables between two groups. The median (interquartile interval (IQR)) of quantitative variables with skewed distribution. In addition, numbers (percentages) were used to represent qualitative variables. Log-rank test was used to compare the time to deterioration, and univariate and multivariate Cox regression analysis was used to screen for factors related to quality of life in ESCC patients. The statistical analysis described above was completed using SPSS version 22.0. The QoLR package of R software was used to develop the TTD model for evaluating scores on the EORTC QLQ-C30/EORTC QLQ-OES18 scale. A 95% confidence interval (95% CI) was used to estimate hazard ratios (HR). Statistical tests were conducted on a two-sided basis with a significance level of 5%, and results with a *p*-value of less than 0.05 were regarded as statistically significant.

## 3. Results

### 3.1. Association with Demographic and Clinical Characteristics

The distribution of demography and clinical characteristics between the high DAI group and the low DAI group was analyzed using the chi-square test. The percentage of males (74.3%) in the low DAI group was greater than that observed in the high DAI group, and the remaining variables were evenly distributed in the two groups (Table 1).

### 3.2. Baseline Quality-of-Life Scores

Baseline quality of life scores were described using measures such as median and quartiles. Statistical analysis showed that the baseline scores in the emotional functioning, appetite loss, and choking when swallowing domain were significantly different between the two groups (*p* < 0.05), no differences in other domains (Table 2). 

### 3.3. Effects of Preoperative DAI on Postoperative Quality of Life in Patients with ESCC

At each follow-up time point, the number of patients experiencing deterioration in each domain was counted, as well as the percentage of all deteriorating patients in that particular domain (Appendix A). The number of quality-of-life deterioration events was evenly distributed between the high and low DAI groups, there was no statistically significant difference in the incidence of deterioration events between the two groups (Appendix A). Log-rank test was utilized to analyze the time of deterioration on the EORTC QLQ-C30/EORTC QLQ-OES18 scale in ESCC patients, and a comparison was made between patients with low and high DAI, compared to low DAI, high DAI could delay the deterioration of global health status (*p* = 0.043), cognitive functioning (*p* = 0.031), dry mouth (*p* = 0.019), and speech problems (*p* = 0.030) (Table 3). Univariate and multivariate Cox regression analysis results revealed that a positive correlation between preoperative high DAI and global health status (*HR* = 0.718, *95% CI*: 0.532–0.969), cognitive functioning (*HR* = 0.641, *95% CI*: 0.450–0.913), dry mouth (*HR* = 0.637, *95% CI*: 0.445–0.911), and speech problems (*HR* = 0.651, *95% CI*: 0.449–0.945) (Table 4, Figure 1).

## 4. Discussion

Assessment of the quality of life of individuals with cancer is increasingly becoming a prominent topic in the literature and expert conversations. Furthermore, it is becoming a key criterion for evaluating the efficacy of tumor treatments. Quality of life involves utilizing individualized and subjective methods toward patients and enables the evaluation of how disease and treatment affect the physical, mental, and social well-being of patients and their relatives [23]. The concept of quality of life is intricate and covers various aspects, including physical activity and psychological well-being, which are closely associated with feelings of happiness, satisfaction, and fulfillment. Health-related quality of life is considered an important endpoint in cancer clinical trials in recent years and is considered a priority area by oncologists [24]. 

In our study, we conducted approximately 7 years follow-up for the patient after surgery and it was found that intake of dietary antioxidants before the surgery can delay and improve the deterioration of postoperative health-related quality of life in ESCC patients, including global health status, cognitive functions, dry mouth, and speech problems, which provides beneficial guidance for improving patients’ postoperative life happiness. According to our research results, it is recommended that patients increase their intake of dietary antioxidants before surgery, such as daily intake of fruits and vegetables. Obviously, this is more beneficial for patients who can undergo selective surgery.

Previous studies reported that consuming foods or dietary supplements rich in antioxidants can enhance the body’s antioxidant defense ability and prevent and slow down the progression of many chronic diseases and diseases [25]. The relationship between Mediterranean diet (MD) and human health was well demonstrated. Mediterranean diet is a typical high dietary antioxidant diet, rich in anti-inflammatory and antioxidant components that help improve reproductive health, reduce the risk of neurodegenerative diseases, and prevent diseases such as depression and psychosocial disorders, which is beneficial for metabolic health and overall wellbeing [26]. In addition, a Chinese study [27] found that the dietary intake pattern of high antioxidant diet before diagnosis can improve the quality of life of breast cancer patients in many aspects, including global health status/QoL and physical function et al. To sum up, these findings support that our research results are more reliable and persuasive.

In the public perception, cancer is a fatal disease, and most patients may experience severe anxiety and depression after cancer diagnosis [28]. During the treatment process, symptoms such as cancer-related cognitive impairment may occur [29]. As demonstrated by previous studies, patients with esophageal cancer commonly have cognitive impairment after surgery, especially esophageal squamous cell carcinoma, which is significantly related to their adverse health outcomes [30]. Kalt W et al. [31] showed that the improvement of cognitive function by dietary antioxidants is mainly related to nutritional nerves. They pointed out that blueberry supplement intervention can reduce inflammation and oxidative stress markers, and upregulate the mechanisms of human and animal neurogenesis, neuroplasticity, and brain-derived neurotrophic factor to protect brain regions, by improving cognitive function. Meanwhile, Saini RK [32] et al. directly pointed out that the antioxidant and anti-inflammatory activities of dietary carotenoids can prevent cognitive decline and neurodegenerative diseases. The above studies were one piece of evidence for the results of our study, providing support for our research findings. Therefore, our study found that increasing dietary antioxidant intake before the diagnosis can improve cognitive function after ESCC surgery, which is consistent with previous research findings and has significant implications for reducing the occurrence of adverse health consequences in patients after surgery.

Patients who underwent esophagectomy often have a variety of long-term persistent symptoms, such as reflux and eating problems. Minimally invasive surgical methods can reduce reflux problems [33], but symptoms related to reflux are difficult to treat and difficult to improve through surgical changes, which is crucial to find other improvement methods. Zhang et al. found a negative correlation between dietary antioxidant vitamin intake and esophageal reflux [34]. Although no effect on reflux symptoms was observed in this study, we found that the reflux-related symptoms (dry mouth) could be improved, which was consistent with other studies [35]. Therefore, we speculated that consuming an appropriate amount of food with high antioxidant activity before the diagnosis can help consolidate the endogenous antioxidant system [36], and improves reflux and reflux-related symptoms (dry mouth). However, the specific mechanism still needs further research and verification.

Diet planning plays a very important role in cancer treatment. A proper diet plan can help patients maintain good health while also reducing discomfort and side effects of treatment [37]. Antioxidants play an important role in cancer treatment, and proper intake can help patients maintain a normal weight, strengthen the body’s resistance, reduce the risk of infection, and improve the success of treatment [38], which supports our findings. Hence, the diet plan should include foods rich in these nutrients, such as fruits, vegetables, and nuts. Additionally, earlier research [39] indicated that the presence of antioxidants in the diet can help govern the structure and operation of the gut microbiome, preserve balance within the intestines, and manage the immune system and inflammatory reactions of the host. Therefore, regulating gut microbiota through the diet may be helpful in cancer treatment. To sum up, a well-designed clinical diet plan is crucial in the treatment of cancer, which can promote cancer treatment and recovery through different mechanisms.

This study had several limitations that should be considered. One of the limitations of this study was that the sample size was relatively small, which may have limited the generalizability of the results. Fortunately, the patients were highly compliant, which allowed us to collect multiple follow-up pieces of information over a period of approximately 7 years after surgery. This enabled us to evaluate the long-term impact of pre-diagnosis dietary antioxidant intake on the quality of life of patients with ESCC after surgery from a longitudinal perspective. Secondly, dietary information was not collected during follow-up, making it difficult to accurately assess postoperative dietary intake. However, the patients we included were all residents of Fujian for more than ten years, with relatively fixed dietary habits. One of the advantages of this study was the long follow-up time, which allowed for the observation of the changing trends in various domains of quality of life during this period; Secondly, a comprehensive antioxidant index was used to comprehensively evaluate the antioxidant content of the diet. Finally, the time to deterioration model was used to calculate the quality of life data, which can find the determination of clinically meaningful time to deterioration, so that clinicians can adjust the treatment plan according to the results in a timely manner. 

In summary, increasing the intake of foods with high antioxidant content before diagnosis in patients with esophageal squamous cell carcinoma can improve their health-related quality of life after surgery, including global health status, cognitive functions, dry mouth, and speech problems. Therefore, this study suggests that high-risk individuals with esophageal cancer should pay attention to the intake of antioxidant foods in their daily diet. However, further research is needed to strictly define the threshold for dietary antioxidant intake.

## Figures and Tables

**Figure 1 nutrients-15-02828-f001:**
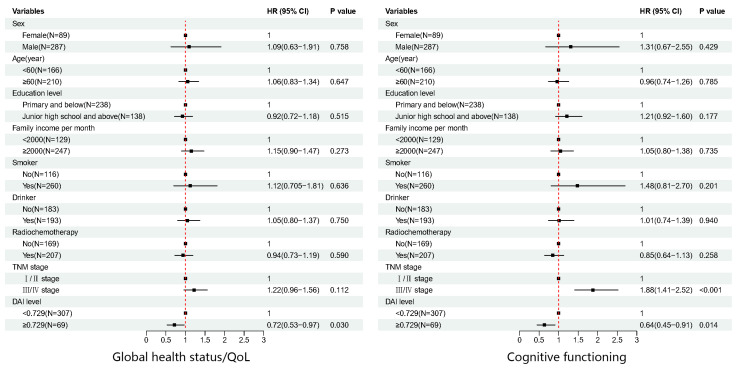
The associations between demographic and clinical characteristics and EORTC QLQ-C30/QLQ-OES18 scales time to deterioration event.

**Table 1 nutrients-15-02828-t001:** Demographic and clinical characteristics of 376 patients with ESCC, stratified by preoperative dietary antioxidant index.

Variables	DAI < 0.729 [*n* (%)]	DAI ≥ 0.729 [*n* (%)]	*χ* ^2^	*p* Value
Sex			3.940	0.047
Female	79 (25.7%)	10 (14.5%)		
Male	228 (74.3%)	59 (85.5%)		
Age (years)			0.901	0.343
<60	132 (43.0%)	34 (49.3%)		
≥60	175 (57.0%)	35 (50.7%)		
Education level			3.405	0.065
Primary and below	201 (65.5%)	37 (53.6%)		
Junior high school and above	106 (34.5%)	32 (46.4%)		
Family income per month			3.507	0.061
<2000	112 (36.5%)	17 (24.6%)		
≥2000	195 (63.5%)	52 (75.4%)		
Smoker			2.326	0.127
No	100 (32.6%)	16 (23.2%)		
Yes	207 (67.4%)	53 (76.8%)		
Drinker			0.912	0.340
No	153 (49.8%)	30 (43.5%)		
Yes	154 (50.2%)	39 (56.5%)		
Radiochemotherapy			2.359	0.125
No	178 (58.0%)	33 (47.8%)		
Yes	129 (42.0%)	52.2%)		
TNM stage			0.651	0.420
I–II	141 (45.9%)	28 (40.6%)		
III–IV	166 (54.1%)	41 (59.4%)		

**Table 2 nutrients-15-02828-t002:** Preoperative EORTC scale baseline scores in male post-ESCC patients.

Domain/Scale	Baseline HRQOL Scores [M (IQR)], *n* = 376			
	DAI < 0.729	DAI ≥ 0.729	Z	*p*
QLQ-C30				
Global health status/QOL	75.00 (66.67, 83.33)	83.33 (66.67, 83.33)	−0.301	0.763
Functional scales				
Physical functioning	100.00 (93.33, 100.00)	100.00 (93.33, 100.00)	−0.702	0.483
Role functioning	100.00 (100.00, 100.00)	100.00 (100.00, 100.00)	−0.603	0.547
Emotional functioning	100.00 (83.33, 100.00)	91.67 (75.00, 100.00)	−2.157	0.031
Cognitive functioning	100.00 (83.33, 100.00)	100.00 (83.33, 100.00)	−0.889	0.374
Social functioning	100 (66.67, 100.00)	91.67 (66.67, 100.00)	−0.493	0.622
Symptom scales				
Fatigue	00.00 (00.00, 22.22)	11.11 (00.00, 22.22)	−0.210	0.833
Nausea/vomiting	00.00 (00.00, 00.00)	00.00 (00.00, 00.00)	−1.107	0.268
Pain	00.00 (00.00, 16.67)	00.00 (00.00, 00.00)	−0.122	0.903
Dyspnea	00.00 (00.00, 00.00)	00.00 (00.00, 00.00)	−0.691	0.490
Insomnia	00.00 (00.00, 00.00)	00.00 (00.00, 33.33)	−1.253	0.210
Appetite loss	00.00 (00.00, 00.00)	00.00 (00.00, 33.33)	−2.339	0.019
Constipation	00.00 (00.00, 00.00)	00.00 (00.00, 00.00)	−0.050	0.960
Diarrhea	00.00 (00.00, 00.00)	00.00 (00.00, 00.00)	−1.267	0.205
QLQ-QES18				
General symptom scales				
Dysphagia	88.89 (66.67, 100.00)	88.89 (77.78, 100.00)	−0.316	0.752
Eating problems	00.00 (00.00, 16.67)	8.33 (0.00, 16.67)	−1.379	0.168
Reflux	00.00 (00.00, 00.00)	00.00 (00.00, 8.33)	−0.494	0.621
Odynophagia	11.11 (00.00, 22.22)	11.11 (00.00, 22.22)	−0.133	0.894
General symptom items				
Trouble swallowing saliva	00.00 (00.00, 00.00)	00.00 (00.00, 00.00)	−0.744	0.457
Choking when swallowing	00.00 (00.00, 33.33)	33.33 (00.00, 33.33)	−2.372	0.018
Dry mouth	00.00 (00.00, 00.00)	00.00 (00.00, 25.00)	−0.603	0.546
Trouble with taste	00.00 (00.00, 00.00)	00.00 (00.00, 00.00)	−1.006	0.286
Coughing	00.00 (00.00, 00.00)	00.00 (00.00, 00.00)	−0.497	0.619
Speech problems	00.00 (00.00, 00.00)	00.00 (00.00, 00.00)	−1.653	0.098

**Table 3 nutrients-15-02828-t003:** Determination of clinically meaningful time to deterioration in the EORTC QLQ-C30/EORTC QLQ-OES18 scale in ESCC patients with low and high DAI.

Domain/Scale	Time to Deterioration [M (95% CI)], *n* = 376			
	DAI < 0.729	DAI ≥ 0.729	*χ* ^2^	*p*
QLQ-C30				
Global health status/QOL	11.072 (9.466–12.678)	13.569 (11.481–15.656)	4.094	0.043
Functional scales				
Physical functioning	11.828 (10.832–12.823)	12.255 (9.369–15.141)	2.198	0.138
Role functioning	14.127 (12.419–15.835)	17.216 (11.712–22.720)	1.951	0.163
Emotional functioning	18.497 (15.124–21.869)	25.823 (19.396–32.251)	3.582	0.058
Cognitive functioning	23.031 (18.392–27.669)	40.378 (19.070–61.685)	4.667	0.031
Social functioning	17.446 (14.956–19.935)	21.848 (11.865–31.831)	1.381	0.240
Symptom scales				
Fatigue	13.109 (11.367–14.851)	14.719 (11.027–18.411)	2.298	0.130
Nausea/vomiting	19.088 (15.517–22.659)	20.534 (11.432–29.636)	0.958	0.328
Pain	20.140 (15.433–24.846)	26.349 (24.340–28.358)	1.509	0.219
Dyspnea	19.844 (16.274–23.414)	24.312 (13.268–35.356)	1.188	0.276
Insomnia	21.158 (17.521–24.796)	20.797 (10.874–30.720)	0.110	0.740
Appetite loss	20.698 (16.461–24.935)	17.183 (5.474–28.891)	0.336	0.562
Constipation	32.657 (28.272–37.042)	26.612 (13.509–39.714)	0.049	0.825
Diarrhea	21.717 (18.311–25.122)	20.172 (10.784–29.560)	0.020	0.887
QLQ-QES18				
General symptom scales				
Dysphagia	12.123 (10.936–13.311)	13.372 (10.808–15.935)	0.751	0.386
Eating problems	14.883 (13.354–16.412)	17.183 (10.807–23.559)	2.861	0.091
Reflux	14.127 (12.147–16.107)	14.653 (12.474–16.832)	0.343	0.558
Odynophagia	20.994 (16.282–25.705)	26.612 (16.847–36.377)	1.357	0.244
General symptom items				
Trouble swallowing saliva	31.014 (24.899–37.130)	37.651 (24.651–50.651)	1.133	0.287
Choking when swallowing	20.534 (15.617–25.450)	27.992 (22.081–33.903)	2.917	0.088
Dry mouth	25.856 (20.067–31.645)	45.667 (15.969–75.365)	5.479	0.019
Trouble with taste	32.526 (27.953–37.458)	40.378 (3.537–77.219)	0.638	0.425
Coughing	23.589 (19.014–28.165)	29.142 (12.518–45.765)	2.254	0.133
Speech problems	30.324 (24.508–36.141)	45.667 (32.126–59.209)	4.685	0.030

**Table 4 nutrients-15-02828-t004:** Association between preoperative DAI and EORTC QLQ-C30/EORTC QLQ-OES18 scale in ESCC patients.

Domain/Scale	Univariate		Multivariate	
	*HR* (*95% CI*)	*p* Value	*HR* (*95% CI*) *	*p* Value
QLQ-C30				
Global health status/QOL	0.739 (0.550–0.992)	0.044	0.718 (0.532–0.969)	0.030
Physical functioning	0.807 (0.607–1.073)	0.139	0.771 (0.737–2.110)	0.077
Role functioning	0.807 (0.598–1.091)	0.164	0.809 (0.595–1.100)	0.176
Emotional functioning	0.727 (0.522–1.013)	0.060	0.717 (0.514–1.001)	0.051
Cognitive functioning	0.682 (0.480–0.967)	0.032	0.641 (0.450–0.913)	0.014
Social functioning	0.827 (0.601–1.136)	0.241	0.846 (0.612–1.169)	0.310
Fatigue	0.790 (0.582–1.073)	0.131	0.779 (0.572–1.061)	0.113
Nausea/vomiting	0.856 (0.628–1.169)	0.329	0.877 (0.639–1.204)	0.417
Pain	0.819 (0.594–1.127)	0.220	0.750 (0.541–1.040)	0.085
Dyspnea	0.834 (0.601–1.157)	0.277	0.836 (0.599–1.167)	0.292
Insomnia	0.949 (0.698–1.291)	0.741	0.925 (0.676–1.265)	0.625
Appetite loss	0.912 (0.669–1.245)	0.563	0.897 (0.655–1.229)	0.499
Constipation	1.040 (0.735–1.470)	0.825	1.000 (0.705–1.418)	1.000
Diarrhea	1.023 (0.745–1.406)	0.887	1.014 (0.734–1.401)	0.934
QLQ-QES18				
Dysphagia	0.877 (0.652–1.181)	0.387	0.882 (0.652–1.194)	0.418
Eating problems	0.765 (0.560–1.045)	0.092	0.757 (0.551–1.039)	0.085
Reflux	0.919 (0.692–1.220)	0.559	0.895 (0.671–1.192)	0.448
Odynophagia	0.822 (0.591–1.144)	0.245	0.790 (0.563–1.108)	0.171
Trouble swallowing saliva	0.826 (0.581–1.175)	0.288	0.776 (0.544–1.108)	0.162
Choking when swallowing	0.744 (0.529–1.046)	0.089	0.768 (0.544–1.084)	0.133
Dry mouth	0.658 (0.462–0.937)	0.020	0.637 (0.445–0.911)	0.014
Trouble with taste	0.862 (0.598–1.242)	0.425	0.878 (0.606–1.272)	0.491
Coughing	0.766 (0.541–1.086)	0.135	0.751 (0.527–1.069)	0.112
Speech problems	0.668 (0.462–0.965)	0.032	0.651 (0.449–0.945)	0.024

* Adjusting for sex, age, education level, family income per month, smoker, drinker, radiochemotherapy, and TNM stage.

## Data Availability

The corresponding author can provide the datasets generated and/or analyzed during the current study upon reasonable request.

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
