# Peer review of "Association between the Preoperative Dietary Antioxidant Index and Postoperative Quality of Life in Patients with Esophageal Squamous Cell Carcinoma: A Prospective Study Based on the TTD Model"

_nutrients, 2023, doi:10.3390/nu15132828_

Round 1
Reviewer 1 Report
The article by Jewei et al''Association between the preoperative dietary antioxidant index and postoperative quality of life in patients with esophageal squamous cell carcinoma: a prospective study based on the TTD model". The study investigates the effects of preoperative dietary antioxidant index (DAI) on health-related quality of life in patients with esophageal squamous cell carcinoma (ESCC), and the results indicate that high DAI can delay the occurrence of postoperative physical function and symptom deterioration, which is positively correlated with the postoperative quality of life.
While going through the manuscript I observed some major comments and few comments limited to language and presentation.
1. The abstract of the article titled "Association between the preoperative dietary antioxidant index and postoperative quality of life in patients with esophageal squamous cell carcinoma: a prospective study based on the TTD model" by Jewei et al. requires revision. Currently, it appears disconnected and lacks essential information regarding the dietary antioxidant index (DAI), the rationale behind the study, and the background information necessary to understand its significance.
2. It is recommended to include additional evidence and references to support the claim of a negative correlation between DAI and cancer risk. This would enhance the credibility of the study and provide readers with a more comprehensive understanding of the topic.
3. It would be valuable to know whether the authors observed any correlations between the preoperative and postoperative periods. This information would help establish a clearer understanding of the longitudinal effects of DAI on patient outcomes and provide insights into the potential benefits of maintaining a high DAI throughout the entire treatment process.
4. As the study is limited to Fijian hospitals and the Fijian population, it would be informative to investigate whether there are any differences in the observed effects of DAI based on demographic variations within the population. This consideration would provide valuable insights into potential variations in response to DAI among different groups, helping to identify any potential disparities in treatment outcomes.
5. Figure 1 appears to be of poor quality, making it difficult to discern the details presented. It is essential to improve the image quality to ensure that readers can adequately comprehend the information conveyed. Clear, legible figures are crucial for effectively communicating research findings.
6. It is recommended to include a paragraph discussing the clinical relevance of the proposed diet plan in relation to cancer treatment. This would elucidate how the findings of the study could potentially impact clinical practice and contribute to the overall management of esophageal squamous cell carcinoma.
The language used in the article is generally acceptable; however, there are minor errors that can be rectified during the proofreading process. These errors do not significantly hinder the understanding of the content but can be addressed to enhance the overall quality of the manuscript.
Author Response
Response to Reviewer 1 Comments
Reviewer
Comments and Suggestions for Authors
The article by Jewei et al''Association between the preoperative dietary antioxidant index and postoperative quality of life in patients with esophageal squamous cell carcinoma: a prospective study based on the TTD model". The study investigates the effects of preoperative dietary antioxidant index (DAI) on health-related quality of life in patients with esophageal squamous cell carcinoma (ESCC), and the results indicate that high DAI can delay the occurrence of postoperative physical function and symptom deterioration, which is positively correlated with the postoperative quality of life.
While going through the manuscript I observed some major comments and few comments limited to language and presentation.
Point 1: The abstract of the article titled "Association between the preoperative dietary antioxidant index and postoperative quality of life in patients with esophageal squamous cell carcinoma: a prospective study based on the TTD model" by Jewei et al. requires revision. Currently, it appears disconnected and lacks essential information regarding the dietary antioxidant index (DAI), the rationale behind the study, and the background information necessary to understand its significance.
Responses 1:
Thank you very much for your advice. We have supplemented background information on dietary antioxidant index in the objective section of the abstract.
Revisions: The changes are as follows:
Page 1 Line 24 to 29:
Objective: Dietary antioxidants are associated with risk of death in cancer patients, and it has been used to evaluate the prognosis of cancer patients. Dietary antioxidant index (DAI) can be used to evaluate dietary antioxidant content comprehensively, this study aimed to investigate the effect of preoperative DAI on health-related quality of life in patients with esophageal cell squamous carcinoma (ESCC).
Point 2: It is recommended to include additional evidence and references to support the claim of a negative correlation between DAI and cancer risk. This would enhance the credibility of the study and provide readers with a more comprehensive understanding of the topic.
Responses 2:
Thank you very much for your insightful comment. We have supplemented more evidence and references in the introduction section to prove that DAI is negatively correlated with cancer risk.
Revisions: The changes are as follows:
Page 3 Line 134 to 137:
Previous studies have found a negative correlation between DAI and cancer risk, such as gastric cancer (Vahid et al. 2020), breast cancer (Vahid et al. 2023), and lung cancer (Wright et al. 2004).
References
Wright, Margaret E.; Mayne, Susan T.; Stolzenberg-Solomon, Rachael Z.; Li, Zhaohai; Pietinen, Pirjo; Taylor, Philip R. et al. (2004): Development of a comprehensive dietary antioxidant index and application to lung cancer risk in a cohort of male smokers. In American journal of epidemiology 160 (1), pp. 68–76. DOI: 10.1093/aje/kwh173.
Vahid, Farhad; Rahmani, Diana; Davoodi, Seyed Hossein (2020): Validation of Dietary Antioxidant Index (DAI) and investigating the relationship between DAI and the odds of gastric cancer. In Nutrition & Metabolism 17. DOI: 10.1186/s12986-020-00529-w.
Vahid, Farhad; Rahmani, Wena; Khodabakhshi, Adeleh; Davoodi, Sayed Hossein (2023): Associat-ed between Dietary Antioxidant Index (DAI) and Odds of Breast Cancer and Correlation be-tween DAI with Pathobiological Markers: Hospital-Based Incidence Case-Control Study. In Journal of the American Nutrition Association 42 (4), pp. 386–392. DOI: 10.1080/07315724.2022.2056543.
Point 3: It would be valuable to know whether the authors observed any correlations between the preoperative and postoperative periods. This information would help establish a clearer understanding of the longitudinal effects of DAI on patient outcomes and provide insights into the potential benefits of maintaining a high DAI throughout the entire treatment process.
Responses 3:
Thank you for your valuable comments. Our study only collected dietary information from patients over the past year before surgery, and did not continue to follow up with dietary information from patients after surgery or discharge. This is also one of the limitations of our study, as described in our discussion. We consider improving this aspect in future studies and strive to collect dietary information from patients at different treatment stages to better observe the effects of dietary antioxidants on patient outcomes in a longitudinal manner.
Point 4: As the study is limited to Fijian hospitals and the Fijian population, it would be informative to investigate whether there are any differences in the observed effects of DAI based on demographic variations within the population. This consideration would provide valuable insights into potential variations in response to DAI among different groups, helping to identify any potential disparities in treatment outcomes.
Responses 4:
Thank you very much for your valuable suggestions. Our study was only conducted in the Fujian region and cannot obtain the dietary characteristics of the population in other regions, which is one of the limitations of our research.
Point 5: Figure 1 appears to be of poor quality, making it difficult to discern the details presented. It is essential to improve the image quality to ensure that readers can adequately comprehend the information conveyed. Clear, legible figures are crucial for effectively communicating research findings.
Responses 5:
Thank you very much for your suggestion. We have improved the image quality of Figure 1.
Point 6: It is recommended to include a paragraph discussing the clinical relevance of the proposed diet plan in relation to cancer treatment. This would elucidate how the findings of the study could potentially impact clinical practice and contribute to the overall management of esophageal squamous cell carcinoma.
Responses 6:
Thank you very much for your advice. We have supplemented a paragraph to the discussion about the relationship between clinically relevant diet plans and cancer treatment.
Revisions: The changes are as follows:
Page 12 Line 386 to 403:
Diet planning plays a very important role in cancer treatment. A proper diet plan can help patients maintain good health while also reducing discomfort and side effects of treatment (O'Callaghan et al. 2023). Antioxidants play an important role in cancer treatment, and proper intake can help patients maintain a normal weight, strengthen the body's resistance, reduce the risk of infection, and improve the success of treatment (Mohseni et al. 2022), which supports our findings. Hence, the diet plan should include foods rich in these nutrients, such as fruits, vegetables, and nuts. Besides, Earlier research (Dacrema et al. 2022) has indicated that the presence of antioxidants in the diet can help govern the structure and operation of the gut microbiome, preserve balance within the intestines, and manage the immune system and inflammatory reactions of the host. Therefore, regulating gut microbiota through the diet may be helpful in cancer treatment. To sum up, a well-designed clinical diet plan is crucial in the treatment of cancer, which can promote cancer treatment and recovery through different mechanisms.

Reviewer 2 Report
My opinion:
1. very intersting topic but only 376 participants
2. a well-chosen diet containing antioxidants
3. observation period too short (7 years)
Author Response
Response to Reviewer 2 Comments
Reviewer
Comments and Suggestions for Authors
My opinion:
Point 1: very intersting topic but only 376 participants
Responses 1:
Thank you very much for your valuable suggestions. One of the limitations of our research is that the sample size is too small, but fortunately, the patients were highly compliant, which allowed us to collect multiple follow-up pieces of information over a period of approximately 7 years after surgery. This enabled us to evaluate the long-term impact of pre-diagnosis dietary antioxidant intake on the quality of life of patients with ESCC after surgery from a longitudinal perspective.
Point 2: a well-chosen diet containing antioxidants
Responses 2:
Thank you very much for your insightful suggestion. The dietary antioxidant index that we use in our research is proposed by Wright [1], which includes six different antioxidants in the diet: vitamins A, C, and E, manganese, zinc, and selenium. The intake of each of these six antioxidants per person is normalized and then added together to obtain the dietary antioxidant index. Currently, this index has been widely used in research. For example, Zhao et al. [2] studied the relationship between dietary antioxidant index and depression; Wang et al. [3] studied the association of the Composite dietary antioxidant index with all-cause and cardiovascular mortality; Wang et al. [4] studied the association between composite dietary antioxidant index and handgrip strength in American adults. Many studies have also used Wright 's proposed dietary antioxidant index, proving its reliability.
Point 3: observation period too short (7 years)
Responses 3:
Thank you for your valuable comments. In future studies, we will continue to follow up these patients and strive to obtain longer observation results.
References
[1] Wright, M.E.; Mayne, S.T.; Stolzenberg-Solomon, R.Z.; Li, Z.; Pietinen, P.; Taylor, P.R.; Virtamo, J.; Albanes, D. Development of a comprehensive dietary antioxidant index and application to lung cancer risk in a cohort of male smokers. Am. J. Epidemiol. 2004, 160, 68–76, doi:10.1093/aje/kwh173.
[2] Zhao L, Sun Y, Cao R, Wu X, Huang T, Peng W. Non-linear association between composite dietary antioxidant index and depression. Front Public Health. 2022 Oct 13;10:988727. doi: 10.3389/fpubh.2022.988727.
[3] Wang L, Yi Z. Association of the Composite dietary antioxidant index with all-cause and cardiovascular mortality: A prospective cohort study. Front Cardiovasc Med. 2022 Oct 4;9:993930. doi: 10.3389/fcvm.2022.993930.
[4] Wu D, Wang H, Wang W, Qing C, Zhang W, Gao X, Shi Y, Li Y, Zheng Z. Association between composite dietary antioxidant index and handgrip strength in American adults: Data from National Health and Nutrition Examination Survey (NHANES, 2011-2014). Front Nutr. 2023 Mar 31;10:1147869. doi: 10.3389/fnut.2023.1147869.

Reviewer 3 Report
Dear authors and editor
The presented manuscript "Association between the preoperative dietary antioxidant index and postoperative quality of life in patients with esophageal squamous cell carcinoma: a prospective study based on the TTD model" is of limited interest to the scientific community. It presents well-known facts about the role of antioxidants in the development of tumor processes. The authors did a great job, but the data presented are more popular science, and for publication in a scientific journal, more scientific arguments must be provided. For example, about specific antioxidants, their content in certain foods, and specific recommendations for eating.
Author Response
Response to Reviewer 3 Comments
Reviewer
Comments and Suggestions for Authors
Dear authors and editor
The presented manuscript "Association between the preoperative dietary antioxidant index and postoperative quality of life in patients with esophageal squamous cell carcinoma: a prospective study based on the TTD model" is of limited interest to the scientific community. It presents well-known facts about the role of antioxidants in the development of tumor processes. The authors did a great job, but the data presented are more popular science, and for publication in a scientific journal, more scientific arguments must be provided. For example, about specific antioxidants, their content in certain foods, and specific recommendations for eating.
Responses:
Thank you very much for your insightful comment. In our study, a semi-quantitative food frequency questionnaire (FFQ) was used to investigate the frequency and intake of various foods in the past 1 year in the study subjects to obtain the long-term food consumption patterns and dietary habits of individuals. A total of 185 food items were included in 12 categories, including staple foods, potatoes, eggs, soy products, animal meat, fish, shrimp, crab and shellfish, vegetables, fruits and dairy. Subsequently, nutrients were calculated according to the Chinese Food Composition Table (6th edition) based on the nutrient composition of each food. We have provided the specific levels of antioxidants in each type of food in our survey based on the Chinese Food Composition Table (6th edition), as detailed in Table 1. According to our research results, it is recommended that patients increase their intake of dietary antioxidants before surgery, such as daily intake of fruits and vegetables, which is more beneficial for patients who can undergo Elective surgery. We have supplemented the content of dietary suggestions to the discussion.
Revisions: The changes are as follows:
Page 11 Line 322 to 325:
According to our research results, it is recommended that patients increase their intake of dietary antioxidants before surgery, such as daily intake of fruits and vegetables. Obviously, this is more beneficial for patients who can undergo selective surgery.
Table 1. Specific Content of Antioxidants in the Chinese Food Composition Table (6th edition)
Food |
Standard parts in grams |
Antioxidants (mg) |
Rice |
100.00 |
1.90 |
White rice porridge |
50.00 |
0.74 |
Rice noodles |
50.00 |
0.89 |
Cellophane noodles |
50.00 |
3.81 |
Burnt rice at the bottom of a pot |
50.00 |
0.00 |
Noodles |
50.00 |
1.27 |
Instant noodles |
50.00 |
14.62 |
Macaroni |
50.00 |
8.02 |
Steamed bread |
100.00 |
10.59 |
Cereal |
50.00 |
13.33 |
Meat bun |
100.00 |
4.31 |
Dumpling |
100.00 |
11.30 |
Dumpling |
100.00 |
11.30 |
Dumpling |
100.00 |
11.30 |
Dumpling |
100.00 |
11.30 |
Bread |
100.00 |
5.93 |
Oil cake |
100.00 |
26.00 |
Deep-fried dough sticks |
100.00 |
13.06 |
Fried cake |
100.00 |
19.53 |
Sesame seed cake |
100.00 |
12.91 |
Flat meat |
100.00 |
11.30 |
Meat Swallow |
100.00 |
11.30 |
Fish ball |
100.00 |
15.84 |
Cuttlefish balls |
100.00 |
15.84 |
Sweet potato |
150.00 |
14.36 |
Taro |
150.00 |
2.90 |
Potato |
150.00 |
15.21 |
Food |
Standard parts in grams |
Antioxidants (mg) |
Yam bean |
150.00 |
14.79 |
Salted duck egg |
100.00 |
36.93 |
Century egg (duck egg) |
100.00 |
36.93 |
Pickled vegetables |
50.00 |
9.43 |
Fermented bean curd |
50.00 |
14.06 |
Bean paste |
50.00 |
13.61 |
Pork floss |
50.00 |
13.27 |
Ham sausage |
50.00 |
13.28 |
Egg |
75.00 |
16.28 |
Duck egg |
100.00 |
22.63 |
Pork |
100.00 |
10.40 |
Pork chops |
100.00 |
18.51 |
Beef |
100.00 |
8.56 |
Mutton |
100.00 |
10.02 |
Rabbit meat |
100.00 |
12.72 |
Chicken |
150.00 |
19.34 |
Duck |
150.00 |
19.34 |
Pork tripe |
100.00 |
15.12 |
Pork liver |
50.00 |
56.31 |
Pig's feet |
100.00 |
7.01 |
Coagulated pig blood used as a food item |
50.00 |
8.45 |
Chicken gizzard |
150.00 |
14.27 |
Chicken wings |
150.00 |
11.84 |
Chicken feet |
150.00 |
11.84 |
Grass carp |
150.00 |
9.62 |
Silver carp |
150.00 |
18.19 |
Crucian |
150.00 |
17.01 |
Food |
Standard parts in grams |
Antioxidants (mg) |
Bass |
150.00 |
36.70 |
Yellow croaker |
150.00 |
36.44 |
Eel |
150.00 |
38.41 |
Sardine |
150.00 |
49.44 |
Herring |
150.00 |
15.12 |
Mackerel |
150.00 |
38.29 |
Spanish mackerel |
150.00 |
53.96 |
Pomfret |
150.00 |
15.12 |
Hairtail |
150.00 |
38.29 |
Mandarin fish |
150.00 |
28.48 |
Levee fish |
150.00 |
81.93 |
Bream |
150.00 |
13.08 |
Trevally |
150.00 |
26.28 |
Flounder |
150.00 |
0.81 |
Ricefield eel |
150.00 |
40.14 |
Squid |
100.00 |
42.36 |
Octopus |
100.00 |
29.32 |
Crab |
150.00 |
89.17 |
Sea shrimp |
100.00 |
60.75 |
Snail |
200.00 |
50.87 |
Clam |
200.00 |
59.56 |
Oyster |
200.00 |
97.72 |
Razor clam |
200.00 |
59.73 |
Mussel |
200.00 |
57.24 |
Jellyfish |
100.00 |
18.66 |
Milk |
250.00 |
1.37 |
Milk powder |
25.00 |
40.52 |
Yogurt |
100.00 |
3.18 |
Food |
Standard parts in grams |
Antioxidants (mg) |
Soymilk |
250.00 |
3.54 |
Breakfast milk |
250.00 |
1.37 |
Cake |
100.00 |
18.97 |
Peanut |
25.00 |
23.55 |
Nut |
25.00 |
38.28 |
Beverages (containing sugar) |
250.00 |
0.00 |
Beverages (sugar free) |
250.00 |
0.00 |
Coffee |
250.00 |
0.00 |
Carbonated beverages |
250.00 |
0.00 |
Dried beans |
25.00 |
16.55 |
Dried beans |
25.00 |
16.55 |
Soybean milk |
250.00 |
1.50 |
Tofu |
100.00 |
7.98 |
Dried bean curd |
50.00 |
23.03 |
Thin sheets of bean curd |
50.00 |
29.61 |
Sliced bean curd |
50.00 |
16.51 |
Oily bean curd |
50.00 |
28.74 |
Soybean sprouts |
100.00 |
10.64 |
Mung bean sprouts |
100.00 |
4.52 |
Beans |
100.00 |
23.37 |
Pea pods |
100.00 |
17.74 |
kidney bean |
100.00 |
10.37 |
White radish tassel |
100.00 |
77.00 |
Carrot tassel |
100.00 |
77.00 |
carrot |
150.00 |
10.55 |
Turnip |
150.00 |
19.31 |
Lotus root |
150.00 |
20.62 |
Bamboo shoot |
150.00 |
6.56 |
Food |
Standard parts in grams |
Antioxidants (mg) |
Cauliflower |
150.00 |
35.12 |
Pakchoi |
150.00 |
65.32 |
Milk cabbage |
150.00 |
65.32 |
Chinese cabbage |
150.00 |
39.09 |
Onion |
25.00 |
9.43 |
Garlic |
25.00 |
12.33 |
Zizania latifolia |
150.00 |
7.26 |
Edible amaranth |
150.00 |
41.22 |
Cabbage |
150.00 |
41.90 |
Leek |
150.00 |
4.49 |
Water spinach |
150.00 |
6.03 |
Spinach |
150.00 |
36.46 |
Leaf mustard |
150.00 |
15.97 |
Horseradish |
150.00 |
15.97 |
Cabbage mustard |
150.00 |
38.10 |
Chepherd's purse |
150.00 |
46.07 |
Celery |
150.00 |
2.25 |
Rape |
150.00 |
0.68 |
Romaine lettuce |
150.00 |
0.22 |
Wax gourd |
100.00 |
16.18 |
Cucumber |
100.00 |
10.12 |
Towel gourd |
100.00 |
4.58 |
Balsam pear |
100.00 |
57.74 |
Pumpkin |
100.00 |
9.11 |
Chayote |
100.00 |
9.56 |
Tomato |
100.00 |
14.63 |
Pepper |
25.00 |
39.00 |
Eggplant |
100.00 |
6.97 |
Food |
Standard parts in grams |
Antioxidants (mg) |
Green pepper |
25.00 |
39.00 |
Edible Fungus |
50.00 |
2.96 |
Flammulina velutipes |
50.00 |
3.88 |
Mushroom |
50.00 |
3.88 |
Lentinus |
50.00 |
3.88 |
Shallot |
25.00 |
23.13 |
Laminaria japonica |
50.00 |
11.62 |
Apple |
250.00 |
3.60 |
Banana |
200.00 |
9.95 |
Orange |
250.00 |
22.24 |
Ggrapefruit |
250.00 |
22.24 |
Pear |
250.00 |
5.91 |
Peach |
200.00 |
11.39 |
Mango |
150.00 |
26.02 |
Pineapple |
150.00 |
19.42 |
Muskmelon |
100.00 |
16.00 |
Grape |
150.00 |
5.17 |
Persimmon |
100.00 |
31.95 |
Longan |
150.00 |
44.30 |
Lychee |
150.00 |
41.40 |
Loquat |
200.00 |
9.51 |
Watermelon |
500.00 |
6.03 |
Strawberry |
100.00 |
49.04 |
Kiwifruit |
100.00 |
66.02 |
Pitaya |
150.00 |
3.65 |
Water chestnut |
100.00 |
8.80 |
Dried shiitake mushrooms |
50.00 |
26.12 |
Dry kelp |
50.00 |
11.62 |
Food |
Standard parts in grams |
Antioxidants (mg) |
Dried seaweed |
50.00 |
17.94 |
Scallops (scallops)) |
50.00 |
57.24 |
Dried fish fillets |
50.00 |
4.36 |

Round 2
Reviewer 1 Report
After carefully considering the comments provided, we are pleased to inform you that the authors have diligently addressed the mentioned remarks. Consequently, we are confident that the article is now suitable for acceptance.
There are minor errors that can be corrected during proof reading
Reviewer 3 Report
Dear Editor and Authors
The authors have done a great job of improving the article. The manuscript may be published.